# Biological Evaluations, NMR Analyses, Molecular Modeling Studies, and Overview of the Synthesis of the Marine Natural Product (−)-Mucosin [note 1]

**DOI:** 10.3390/molecules29050994

**Published:** 2024-02-24

**Authors:** Jens M. J. Nolsøe, Jarl Underhaug, Åshild Moi Sørskar, Simen Gjelseth Antonsen, Karl E. Malterud, Osman Gani, Qiong Fan, Marit Hjorth, Thomas Sæther, Trond V. Hansen, Yngve H. Stenstrøm

**Affiliations:** 1Faculty of Chemistry, Biotechnology and Food Science, Norwegian University of Life Sciences, P.O. Box 5003, NO-1433 Ås, Norway; jens.m.nolsoe@nord.no (J.M.J.N.); t.v.hansen@farmasi.uio.no (T.V.H.); 2Faculty of Biosciences and Aquaculture, Nord University, P.O. Box 1490, NO-8049 Bodø, Norway; 3Department of Chemistry, University of Bergen, Allégaten 41, NO-5007 Bergen, Norway; jarl.underhaug@uib.no; 4Department of Pharmacy, Section for Pharmaceutical Chemistry, University of Oslo, P.O. Box 1068, NO-0316 Oslo, Norway; a.m.sorskar@farmasi.uio.no (Å.M.S.); k.e.malterud@farmasi.uio.no (K.E.M.); osman.gani@farmasi.uio.no (O.G.); 5Department of Mechanical, Electronic and Chemical Engineering, Faculty of Technology, Art and Design, Oslo Metropolitan University, NO-0130 Oslo, Norway; simenant@oslomet.no; 6Department of Molecular Medicine, Institute of Basic Medical Sciences, Faculty of Medicine, University of Oslo, NO-0317 Oslo, Norway; qiong.fan@outlook.com (Q.F.); marit.hjorth@medisin.uio.no (M.H.); thomas.sather@medisin.uio.no (T.S.)

**Keywords:** natural products, mucosin, NMR studies, structural elucidation, 15-lipoxygenase, peroxisome proliferator activating receptors, arachidonic acid

## Abstract

Natural products obtained from marine organisms continue to be a rich source of novel structural architecture and of importance in drug discovery, medicine, and health. However, the success of such endeavors depends on the exact structural elucidation and access to sufficient material, often by stereoselective total synthesis, of the isolated natural product of interest. (−)-Mucosin (**1**), a fatty acid derivative, previously presumed to contain a rare *cis*-bicyclo[4.3.0]non-3-ene moiety, has since been shown to be the *trans*-congener. Analytically, the fused bicyclic ring system in (−)-**1** constitutes a particular challenge in order to establish its relative and absolute stereochemistry. Herein, data from biological evaluations, NMR and molecular modeling studies of (−)-**1** are presented. An overview of the synthetic strategies enabling the exact structural elucidation of (−)-mucosin (**1**) is also presented.

## 1. Introduction

Polyunsaturated fatty acids (PUFAs) display rather modest structural complexity [1]. However, when not integrated as constituents of the eukaryotic cell membrane or serving as a fuel repository, further enzymatic transformation can result in a plethora of structurally diverse natural products [2,3,4]. Particularly, the marine environment has provided an array of diverse naturally occurring carbocycles [4,5,6], where the prostaglandin family [7] is a classic example. Prostaglandins, such as 15*S*-PGA_2_ (**3**) (Figure 1), belong to the class of proinflammatory lipid mediators [8,9], but more recent research has nuanced the physiological role of the prostaglandins to be context dependent [10,11], including those isolated from marine habitats [12,13]. Often, these marine carbocyclic oxylipins can be related directly to components found in the human inflammatory metabolome. Two examples of less studied marine carbocyclic compounds are (−)-mucosin (**1**) [14] and (−)-dictyosphaerin (**2**) (Figure 1) [15].

The authors of the present paper have been engaged in a successful campaign that ultimately established the correct structure of the marine eicosanoid (−)-mucosin (**1**) by stereocontrolled total synthesis [16,17,18,19], via the originally claimed structure **5** (Figure 2).

Generally, the fused alicyclic ring system of compounds such as **1** and **2** pose a challenge because the relative stereochemistry usually is assigned on the basis of NMR data alone. Spectral crowding in regions of topological relevance may not allow any clear interpretation. For the same reason, very often stereoselective total synthesis of natural products is required in order to accomplish a complete elucidation [20,21]. Considering the generalized structure portrayed by the mucosin scaffold, keeping the appended double bond fixed in an *E*-geometry, the four contiguous stereocenters can be represented by one of 16 stereoisomers (Figure 3).

Herein, these synthetic efforts are outlined together with NMR data of the revised structure (−)-**1**. In addition, results from molecular modeling studies, 15-lipoxygenase (15-LOX) inhibition experiments, cytotoxicity assays and biological evaluations towards the peroxisome proliferator activating receptors (PPARs) α and γ, using stereoselectively prepared (−)-(**1**), are presented.

## 2. Results and Discussion

### 2.1. Overview of Stereoselective Synthesis of (−)-Mucosin *(**1**)* and Stereoisomers

Isolated in 1997 from the Mediterranean Sea sponge *Renierea mucosa*, the original assignment of (−)-mucosin was performed by Casapullo and co-workers on its methyl ester [14] (Figure 3). Thus, subsequent to HRMS and IR analyses, application of various NMR techniques established that the parent C_20_-compound contains a bicyclo[4.3.0]non-3-ene scaffold. As mentioned, there are 16 stereoisomers of the suggested C_20_-compound, considering the four chiral carbons present in the bicyclic core. Based on their analyses, Casapullo and co-workers suggested the structure **5**, with *cis*-geometry at the fused juncture. The topology of the four interconnected points of chirality was based on correlations observed in NOESY and ROESY experiments. The authors determined the *trans*-configuration of C-8 and C-16 according to steric interactions seen between H_2_-7 and H-16 and H-9, which seems reasonable. However, the assignment of the reported *cis*-configuration with respect to the fusion geometry was not described in detail. The suggested structure (−)-**5** was confirmed by Whitby and co-workers as a result of their reported synthesis of its enantiomer [22], since their NMR data corresponded to the ones reported by Casapullo and co-workers [22]. In addition, the synthetic material showed a specific optical rotation of +38.2° (c = 0.8, hexane) that was comparable to the original reported value of −35.5° (c = 0.8, hexane) [14] for the presumed ester (−)-**6**. This provided support that the enantiomer of originally claimed (−)-**5** (Figure 3) was synthesized. As part of our interest in the biogenesis [22] and the synthesis of (−)-mucosin, formation of the *cis*-fused bicyclic system was achieved by [2+2] cycloaddition of dichloroketene and 1,4-cyclohexadiene. In the two following steps, a Büchner–Curtius–Schlotterbeck ring expansion reaction and a zinc mediated hydrodehalogenation furnished meso-ketone **8**. Subsequently, the pivotal desymmetrization of meso-ketone **8** was executed via a Claisen-type reaction using Simpkin’s base, (+)-*bis*[(*R)*-phenyethyl]amine, and methyl cyanoformate at low temperature to provide β-keto ester **9** as a single isomer [16] (Figure 1).

To ensure the *trans*-relationship between the appended side chains on the cyclopentane ring attributed to claimed (−)-mucosin (**5**), the ketone moiety in **9** was transformed to its enol triflate, which was reacted with CuCN and butyl lithium to yield conjugated ester **10**. The use of magnesium in methanol reduced the α,β-double bond in **10**, as a 2:1 mixture of C8 epimers. This epimeric mixture was then equilibrated to the desired diastereomer **11** in the presence of sodium methoxide. Through a few more reactions, the terminal alkyne **14** was formed via **12** and **13**. Finally, a telescoped sequence involving hydrometallation, halodemetallation and a Negishi type cross-coupling was developed. By this, reaction of the *trans*-vinyl iodide derived from **14** with 4-ethoxy-4-oxobutylzinc bromide, in the presence of Pd(PPh_3_)_4_ as the catalyst, yielded the target molecule as depicted in Figure 2.

However, our NMR and the optical rotation data of −9.8° (c = 0.8, hexane) did not match those published by Casapullo and co-workers [14] nor those of Whitby and co-workers [22]. X-ray crystallography was performed on the 3,5-dinitro benzoate ester of the late stage intermediate **12**, confirming the topological relationship displayed by the featured *cis*-bicyclo[4.3.0]non-3-ene scaffold. Thus, we assumed that the stereochemistry of the appended side chains was wrong. Consequently, it was decided to prepare the diastereomer **15** from **9** (Figure 3), having opposite appended topology relative to (−)-(**5**). However, instead of a conjugate reduction reaction used to supply **11**, we now developed a sequence featuring a conjugate addition (BuMgCl, TMSCl, CuI (10 mol%)) on an unsubstituted Michael acceptor motif obtained from β-keto ester **9** in order to furnish the desired diastereomer **16** [17], see Figure 3. This therefore demonstrated a stereodivergent approach.

Again, our NMR and optical rotation data did not match, but once more X-ray crystallography confirmed the depicted stereochemistry for (+)-**16** (+64° (c = 0.8, hexane)), Figure 3. Our assumption was then that the unusual *cis*-fused topology advocated for (–)-mucosin (**1**) was wrong, especially after biosynthetic considerations with the PUFA **4** as substrate [22]. In the data published by Casapullo and co-workers we could not find support for the claimed *cis*-fused geometry [14]. Nor did we find that Whitby and co-workers were able to corroborate this crucial feature [22]. We therefore concluded that the intended sequence did not furnish the enantiomeric methyl ester ***ent*-6** (Figure 4).

A rationale for the misassignment is an epimerization via formation of competing π-allyl complexes during zirconium-mediated co-cyclization (Figure 5). This is indeed confirmed by X-ray crystallography performed by Whitby and co-workers on a model system subsequent to having performed the featured transformation [22]. In contrast to substrate **17**, the model system only contained terminal alkenes as the participating functionalities. Consequently, the difference in steric requirements of an internal alkene relative to a terminal alkene in zirconium-mediated co-cyclization has plausibly acted as a confounding factor leading ***ent*-7**.

Eventually, we performed DFT calculations comparing geometry-optimized structures of the diastereomers depicted in Figure 3 to find the one with the lowest strain [18], that was then selected as our new synthetic target. Relying on the stereospecific Diels–Alder reaction and an enantioselective literature protocol [23], the stereodefined keto-ester **18**, with the *trans*-fused hexahydroindene system, was prepared. Similar reactions as used before yielded the intermediate **19**, where the structure was again confirmed by X-ray analysis. From **19**, the synthesis of the target molecule (−)-mucosin (**1**), see Figure 6, was based on our established protocols outlined in Figure 2 and Figure 3.

Satisfyingly, this time the NMR data and specific optical rotation value did indeed match the data from both Casapullo and co-workers [14] and Whitby [22] and co-workers. In the case of Whitby, there must have been a confounding factor at work, resulting in the mentioned isomerization of **17** under the applied reaction conditions (Figure 5). Moreover, our efforts also underline the importance of making NMR spectral and raw data available, but also emphasize the limitations in each analytical method. Furthermore, the overview presented herein also underscores the importance of stereoselective total synthesis in exact structural assignments of natural products [22].

### 2.2. NMR Studies

#### 2.2.1. Preliminary Considerations

More than 25 years have passed since Casapullo and co-workers presented their NMR data [14]. However, an important consideration when addressing compact alicyclic structures by NMR, such as (−)-mucosin (**1**) and (−)-dictyosphaerin (**2**), is whether the field strength, and therefore also the width of the spectral window, is adequate to discern pertinent resonances or correlations [24]. However, without having any access to the original raw data, it is difficult to assess this juncture, although the erroneous assignment by Casapullo and co-workers were conducted at both 500 and 600 MHz [14].

While the HMBC correlations trace the general outline of (−)-mucosin (**1**) by accounting for each individual ^1^H-^13^C coupling, the description of the NOESY and ROESY experiments was incomplete in the original report [14], as well as in the article published by Whitby and co-workers [22].

#### 2.2.2. Structural Assignment and Discussion of NMR Data

In Figure 4, the absolute and relative stereochemistry of the methyl ester **7** of the target molecule (−)-**1** are presented.

It must be acceded that (−)-mucosin (**1**), even though a small alicycle, is challenging due the four contiguous stereocenters adorning the bicyclo[4.3.0]non-3-ene system. It was therefore decided to acquire the spectroscopic data with as high a field strength as possible. Thus, we analyzed the prepared samples on an 850 MHz instrument. We were able to assign all protons and carbons using coupling patterns combined with 2D NMR, see Figure 4 for carbon numbering and the supporting information for spectra.

We hoped that further analysis of the NMR interactions of H14 and H9 would reveal the true relative configuration of the bicyclic system. Both H14 and H9 would be expected to be ddddd and dddd, respectively, and truly both are revealed as complex multiplets. Even at the high field strength of 850 MHz, coupling constants were impossible to extract. Also, analysis of H16 and H8 should prove the *trans* relation of the side chains. H8 shows as a very complex multiplet (1.59–1.63 ppm), while H8 overlaps with H9 making the extraction of coupling constants impossible, as the difference from these two protons differ by less than 0.01 ppm.

The topologically distinguishing HSQC-hydrogens are closely spaced together as seen in Figure 5. Despite this, we have been able to plainly assign all the shifts in structure **1** through the application of various types of correlation spectroscopy, including HSQC and HMBC (see Appendix A). Of note are the topological ^1^H-^13^C correlations.

Several different 2D experiments were employed. The best results were obtained with 2D HSQC-NOESY with 0.5 s mixing time. In this spectra, cross peaks for the coupling between H15 and H8, can be seen. In the most stable conformation, the distance is calculated to be 2.6 Å, while it is 4.6 Å for the original suggested structure. The latter will not be visible in this type of NMR spectra. Also, correlation between H16 and H18, is in favor of the *trans*-fused system which is the relative configuration for the natural product. However, we were not able to measure couplings between H7/H17 or H8/H16. It is known from the decalin system that the fusion geometry has a substantial impact on the spectroscopic behavior. For comparison, the axial bridgehead hydrogens in *trans*-decalin are located at 0.87 ppm, while the pseudo-equatorial bridgehead hydrogens in *cis*-decalin are located at 1.58 ppm [25]. However, with a large overlap of the chemical shift of the more diagnostic protons, this was difficult to interpret. Nevertheless, the correlation between H16 and both of the protons at H7 clearly indicated the *trans*-configuration of the two side chains. For H14 and H9, we were not able to get any useful information from NOESY-HSQC. Detailed spectral data can be found in the Appendix A.

### 2.3. Docking Studies of (−)-Mucosin *(**1**)* with 15-LOX-2 and PPARγ

PUFAs and their products are known modulators of the enzyme 15-lipoxygenase (15-LOX) [26], a key enzyme in the biosynthesis of anti- and pro-inflammatory lipid mediators [27]. We then decided, due to the few functionalities in (−)-mucosin (**1**), to perform molecular modeling studies, using the structure of 15-LOX-2 [28]. The natural product **1** occupied nearly half of the binding pocket and showed a favorable docking score of −6.34 Kcal/mol, in comparison to the cognate ligand (−4.4 Kcal/mol). However, relative binding energies, ΔG, (calculated as Molecular Mechanics with Generalized Born and Surface Area solvation method, MM/GBSA) for (−)-**1** is less favorable in comparison to the cognate ligand (−44.0 Kcal/mol vs. −55.2 Kcal/mol, respectively), which can be partly attributed to more favorable coulombic interactions with the cognate ligand (−15.9 Kcal/mol) in comparison to the marine natural product (−)-**1** (+40.73 Kcal/mol).

With PPARγ, (−)-**1** showed less favorable docking score (−6.5 Kcal/mol) in comparison to the cognate ligand (−8.2 Kcal/mol), which is also reflected with less favorable ΔG with MM/GBSA (−53.8 Kcal/mol vs. −65 Kcal/mol, respectively). The results from both docking studies are shown in Figure 6, which encouraged biological evaluations.

### 2.4. Biological Evaluations

As of today, no biological evaluations of (−)-mucosin (**1**) have been reported. Of note, this marine natural product is not rich in functional groups. However, since (−)-**1** is a PUFA derivative, we became interested in testing it against the peroxisome proliferator-activated receptor (PPAR) α and γ. The PPARs are ligand-activated nuclear receptors regulating a wide range of physiological processes [29]. These receptors respond to endogenous ligands like fatty acids, fatty acid derivatives or synthetic analogs. Some analogs have entered the drug market for treatment of various metabolic disorders [29,30,31,32,33]. Several lipid-based natural products have been reported as PPAR agonists serving as lead compounds towards developing new anti-diabetic drugs devoid of the adverse side effects of existing PPAR drugs [34,35]. Of relevance for the structure of (−)-mucosin (**1**), prostaglandins exhibit agonistic effects towards PPARs [36]. Against this background and in relation to our interest in developing PPAR-agonists based on natural products [31,32,37], we subjected (−)-**1** to biological evaluations (Figure 7).

No cytotoxic effects of the marine lipid (−)-**1** were observed in the cell viability test assay (Figure 7A) nor in the lactate dehydrogenase (LDH) assay (Figure 7B). Testing (−)-**1** against a panel of human nuclear receptors (Figure 7C), resulted in a weak activation of the reporter gene by PPARs, but not by Liver X Receptors (LXRs) or Retinoic X Receptor α (RXRα). Of note, (−)-**1** did not exhibit any significant agonistic effects against neither PPARα nor PPARγ (Figure 7D). Also, when stimulating the human hepatoma cell line 

Regarding the inhibition against soybean 15-LOX, no inhibition of soybean 15-LOX was observed at the highest tested concentration (75 μM) for (−)-**1** and its methyl ester (−)-**7**.

## 3. Experimental Section

The NMR spectra were acquired at 298 K on an 850 MHz Bruker AVANCE III HD equipped with a TCI CryoProbe (Bruker BioSpin, Billerica, MA, USA). Coupling constants (*J*) are reported in hertz and chemical shifts (δ) are reported in parts per million (ppm), referenced to the residual solvent signal (7.27 ppm for ^1^H and 77.00 ppm for ^13^C).

### 3.1. 1D NMR Data of Methyl Ester of (−)-Mucosin *(**7**)*

^1^H-NMR (850 MHz, CDCl_3_): d 5.68–5.67 (m, 2H), 5.46 (td, 1H, *J* = 7.2, 15.1 Hz), 5.39 (td, 1H, *J* = 6.8, 15.0 Hz), 3.68 (s, 3H), 2.32 (t, 2H, *J* = 7.6 Hz), 2.28–2.25 (m, 1H), 2.23–2.19 (m, 1H), 2.15–2.09 (m, 2H), 2.04 (dt, 2H, *J* = 7.1, 7.1 Hz), 1.75–1.75 (m, 2H), 1.73–1.73 (m, 2H), 1.71 (tt, 2H, *J* = 7.4, 7.4 Hz), 1.63–1.59 (m, 1H), 1.57 (ddd, 1H, *J* = 2.9, 7.3, 12.2 Hz), 1.54–1.51 (m, 1H), 1.45–1.41 (m, 1H), 1.35 (ddd, 2H, *J* = 11.4, 11.4, 11.4 Hz), 1.32–1.26 (m, 3H), 1.24–1.21 (m, 1H), 1.19–1.15 (m, 1H), 1.15–1.10 (m, 2H), 0.90 (t, 3H, *J* = 7.1 Hz).

^13^C-NMR (214 MHz, CDCl_3_): 174.30, 130.43, 129.94, 127.39, 127.27, 52.38, 51.58, 47.33, 42.46, 40.22, 37.10, 36.88, 36.82, 33.56, 32.55, 32.07, 31.73, 30.88, 24.87, 23.08, 14.31.

The 2D NMR data can be found in Appendix A.

### 3.2. Molecular Modeling Experiments

The 3D coordinates of **1** were downloaded from the PubChem database and geometry optimized by ORCA quantum chemistry program [38] using B3LYP/def2-TZVP basis set. The optimized structure was docked into the 15LOX-2 (PDB ID: 4NRE [28], cognate ligand (hydroxyethyloxy)tri-(ethyloxy)octane) using Glide standard precision docking program [39] in Schrodinger package. For PPARγ (PDB ID; 2ZVT [40], the cognate ligand (5*E*,14*E*)-11-oxoprosta-5,9,12,14-tetraen-1-oic acid) was employed.

### 3.3. Cytotoxicity Assays

Cytotoxic effects of (−)-**1** were evaluated in COS-1 cells (ATCC^®^ CRL-1650; LGC Standards GmbH, Wesel, Germany), using the Roche Cytotoxicity Detection Kit (#1164479300, Sigma-Aldrich, St. Louis, MO, USA), measuring lactate dehydrogenase (LDH) leaked from the cells or by the XTT-based In Vitro Toxicology Assay Kit (#TOX2-1 KT, Sigma-Aldrich), measuring reduced metabolic NAD(P)H flux. The cells were maintained in Dulbecco’s modified Eagle’s medium (DMEM; D6546, Sigma-Aldrich) containing penicillin/streptomycin (50 U/mL; 50 μg/mL), 4 mM L-glutamine, and 10% fetal bovine serum (F7524; Sigma-Aldrich), at 37 °C in a humidified atmosphere of 5% CO_2_ in air. Cell confluence never exceeded 80% before subculturing or transfection. Both cytotoxicity assays were run as described by the manufacturer, and absorbance was read at 492/750 nm and 450/690 nm for the LDH and XTT assay, respectively, on a Synergy H1 Hybrid Multi-Mode Microplate Reader (BioTek^®^ Instruments, Winooski, VT, USA).

### 3.4. Luciferase Assays

For the dose-response and specificity assays, COS-1 cells were seeded at 7 × 10^4^ cells/well on 24-well plates. After 24 h cells were transfected with either 0.1 μg of the Gal4-DBD-NR-LBD expression plasmids, 0.2 μg of the 5× UAS-SV40 luciferase reporter, and 0.05 μg of the Renilla Luciferase-coding internal control (pRL-CMV) using Lipofectamin 2000 (Life Technologies, Carlsbad, CA, USA). The plasmid constructs have been described earlier [31,32]. After 5 h the cells were treated with (−)-**1**, pirinixic acid (WY-14643; C7081, Sigma-Aldrich), or rosiglitazone (BRL-49653; Cayman Chemical, Ann Arbor, MI, USA) in DMSO (final conc. 0.1%). After 18 h cells were washed in PBS and lysed in Passive Lysis Buffer (Promega, Madison, WI, USA) and Dual-Luciferase^®^ ReporterAssay System (Promega) was run on a Synergy H1 Hybrid Multi-Mode Microplate Reader (BioTek® Instruments) following the manufacturers protocol. The Firefly Luciferase readings were normalized to the Renilla Luciferase numbers, and data from at least three independent transfection experiments run in duplicate are presented.

### 3.5. Gene Expression in HepG2 Cells: cDNA Synthesis and Real-Time Quantitative PCR

Human HepG2 cells (ATCC-HB-8065; LGC Standards GmbH) were grown in the same DMEM-based media as the COS-1 cells. The cells were incubated with 50 μM (−)-**1** in DMSO or DMSO only (final conc. 0.5%) for 24 h. RNA was isolated with a NucleoSpin RNA mini kit (Cat# 740955; Machery-Nagel, Düren, Germany), according to the manufacturer’s instructions. Reverse transcription of RNA (500 ng) into cDNA was done using MultiScribe Reverse Transcriptase (Cat# 4311235, Thermo Fisher Scientific, Waltham, MA, USA) and random hexamer primers. Gene expression was measured with RT-qPCR using SsoAdvanced Universal SYBR green Supermix (Cat# 1725271; Bio-Rad, Irvine, CA, USA) on a Bio-Rad CFX96 Touch™. The RT-PCR primers were designed with Primer-BLAST (NCBI, Bethesda, MD, USA) [41], and gene expression was normalized against the expression of TATA-binding protein (*TBP*). Primer sequences are displayed in Table 1.

### 3.6. 15-LOX Inhibition Experiment

Soybean 15-lipoxygenase activity was measured as previously described [42], in borate buffer solutions (0.2 M, pH 9.00) by the increase in absorbance at 234 nm during 30 to 90 s after the addition of the enzyme, using linoleic acid (134 μM) as substrate. The final enzyme concentration was 167 U/mL. Test substances were added as DMSO solutions (final DMSO concentration 1.6%); DMSO alone was added in uninhibited control experiments. Six or more parallels of controls and three parallels of (−)-**1** and (−)-**7** were measured. To ensure constant enzyme activity throughout the experiment, the enzyme solution was kept on ice, and controls were measured at regular intervals. Calculation of enzyme activity was carried out as previously described [42].

### 3.7. Statistical Analysis

Statistical analyses were performed using GraphPad Prism 9 (GraphPad Software Inc., San Diego, CA, USA). All data are presented as mean and standard error of the mean (SEM) or standard deviation (SD). Statistical differences between groups were determined by one-way analysis of variance (ANOVA) followed by Tukey’s multiple comparison tests. For all statistical tests *p* < 0.05 was considered statistically significant.

## 4. Conclusions

In our synthetic approaches towards the true structure of the molecule (−)-mucosin (**1**), a lot of data was obtained [16,17,18,19]. We used an 850 MHz NMR instrument to analyze the correlations between C9 and C14. However, these experiments gave inconclusive results, but the *trans* relationship between C8 and C16 on the cyclopentane ring was confirmed. For the first time using results from biological evaluations for the first time using (−)-mucosin (**1**) have been presented. These showed no cytotoxic effects in the cell viability test assay or in the lactate dehydrogenase assay. Moreover, the lack of inhibition against 15-LOX and potent agonism towards PPARα and PPARγ, are most likely due to the hydrocarbon nature of (−)-mucosin (**1**).

## Data Availability

Data are contained within the article and Appendix A.

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
