# Peer review of "Biological Evaluations, NMR Analyses, Molecular Modeling Studies, and Overview of the Synthesis of the Marine Natural Product (−)-Mucosin"

_molecules, 2024, doi:10.3390/molecules29050994_

Round 1

Reviewer 1 Report

Comments and Suggestions for Authors

The manuscript molecules-2880634, of the title Biological evaluations, NMR analyses, molecular modelling studies, and overview of the synthesis of the marine natural product (-)-mucosinby Nolsøe et al., describes their stereoselective synthesis and revised structural assignment of (-)-mucosin. 

(-)-Mucosin (1) is a polyunsaturated fatty acid (PUFA – arachidonic acid) natural product derived from the marine environment. It was previously believed to possess an unusual cis-bicyclo[4.3.0]non-3-ene moiety, that has since been challenged and shown to be the trans-congener by the authors of the submitted paper. 

Herein, the authors provide an excellent overview of their synthetic strategies enabling the exact structural elucidation of (-)-mucosin. Their effort is strongly flavoured and supported by X-ray crystallography studies, molecular modelling, and last but not least, a highly impressive 850 MHz NMR spectroscopy offering various types of correlation, including HSQC and HMBC, and very noticeable topological 1H-13C correlations. 

In addition, the authors present results from molecular modelling studies, 15-lipoxygenase (15-LOX) inhibition experiments, cytotoxicity assays and biological evaluations towards the peroxisome proliferator activating receptors (PPARs) a and g, using their stereoselectively prepared (-)-mucosin (1). 

This work nicely highlights the importance of a proper structural assignment of synthetic natural products prior to their further screening for the potential biological properties they may offer.

This is a very well and clearly written paper with very well performed studies. I believe the paper is very suitable for publication in Molecules and strongly recommend it to be accepted after only very minor amendments. I have the following couple of minor additional comments: 

1.   Page 5, line 154: I was wondering whether ent-7 in this line rather was supposed to refer to ent-6 (as in Scheme 4). 

2.   Page 7, line 175: the stereodefined keto ester 18, please place the compound number 18 into Scheme 6 accordingly (the number is missing there). 

Author Response

  1. Page 5, line 154: I was wondering whether ent-7 in this line rather was supposed to refer to ent-6 (as in Scheme 4). This has een changed in accord with the comment.
  2. Page 7, line 175: the stereodefined keto ester 18, please place the compound number 18into Scheme 6 accordingly (the number is missing there).  This has been added as suggested.

Reviewer 2 Report

Comments and Suggestions for Authors

The paper "Biological Evaluations, NMR Analyses, Molecular Modelling Studies……“ submitted by Yngve Stenstrom et al. deals with natural product chemistry based on marine organism and provides a solid piece of work for both, synthetic chemists and medicinal chemistry. In addition, the studies could solve an important stereochemical issue establishing the relative and absolute stereochemistry  of the marine natural product (-)-mucosin. Besides that, highly sophisticated NMR studies and molecular modelling data underlined the lab experiments. Finally, the account presents a comprehensive biological evaluation of the material.

In summary, the paper offers very useful information for the the natural product community, it is highly interdisciplinary showing solid data also for the medicinal chemistry interface.

It should be published in Molecules.

Author Response

Thank you very much for taking the time to review this manuscript and for your comments.